# Distilling KGE black boxes into interpretable NeSy models

**Rodrigo Castellano Ontiveros**                    RODRIGO.CASTELLANO@UNISI.COM
*University of Siena, Italy*

**Francesco Giannini**                    FRANCESCO.GIANNINI@SNS.COM
*Scuola Normale Superiore, Pisa, Italy*

**Michelangelo Diligenti**                    MICHELANGELO.DILIGENTI@UNISI.COM
*University of Siena, Italy*

**Editors:** Leilani H. Gilpin, Eleonora Giunchiglia, Pascal Hitzler, and Emile van Krieken

## Abstract

Knowledge Graph Embedding (KGE) models have shown remarkable performances in the knowledge graph completion task, thanks to their ability to capture and represent complex relational patterns. Indeed, modern KGEs encompass different inductive biases, which can account for relational patterns like reasoning compositional chains, symmetries, anti-symmetries, hierarchical patterns, etc. However, KGE models inherently lack interpretability, as their generalization capabilities are purely focused on mapping human interpretable units of information, like constants and predicates, into vector embeddings in a dense latent space, which is completely opaque to a human operator. On the other hand, different Neural-Symbolic (NeSy) methods have shown competitive results in knowledge completion tasks, but their focus on achieving high accuracy often leads to sacrificing interpretability. Many existing NeSy approaches, while inherently interpretable, resort to blending their predictions with opaque KGEs to boost performance, ultimately diminishing their explanatory power. This paper introduces a novel approach to address this limitation by applying a post-hoc NeSy method to KGE models. This strategy ensures both high fidelity to KGE models and the inherent interpretability of NeSy approaches. The proposed framework defines NeSy reasoners that generate explicit logic proofs using predefined or learned rules, ensuring transparent and explainable predictions. We evaluate the methodology using both accuracy and explainability-based metrics, demonstrating the effectiveness of our approach.

## 1. Introduction

Knowledge Graphs (KGs) are structured representations of factual knowledge in the form of interconnected triplets (subject, predicate, object), which have become essential for various real-world AI applications. KGs are inherently incomplete and link prediction is the task of uncovering missing relationships based on the existing graph structure. Knowledge Graph Embedding (KGE) models have emerged as powerful tools for link prediction thanks to their ability to capture complex relational patterns (Wang et al., 2021; Rossi et al., 2021). These models leverage diverse inductive biases to support different inference patterns, such as compositional chains, symmetries, anti-symmetries, and hierarchical patterns (Pavlović and Sallinger, 2023a,b). For this reason, KGEs are perfectly capable of making logical deductions via inference patterns based on the information already present in the graph (Abboud et al., 2020).

However, despite their capabilities, KGE models often suffer from a lack of transparency. Indeed, their reliance on dense vector embeddings in latent spaces renders their reasoning

process opaque to human understanding (Lecue, 2020). While some KGEs may implicitly encode logical properties, like transitivity or symmetry, it remains challenging to discern whether (and which of) these properties have been used in the prediction process of any query (Gutierrez Basulto and Schockaert, 2018; Zhang et al., 2019; Pavlović and Sallinger, 2023a). This opacity hinders trust and limits the adoption of KGE models in critical applications, where explainability is of prime importance.

To address this challenge, various explainability methods for KGEs have been proposed. On one hand, symbolic methods based on rule mining techniques, like AMIE+ (Galárraga et al., 2015), AnyBURL (Meilicke et al., 2019), SAFRAN (Ott et al., 2021) and GenI (Amador-Domínguez et al., 2023), extract logic rules from KGs, often relying on heuristics and generating a vast number of potential rules. While these methods offer insights into the underlying reasoning, they are based on an exhaustive rule mining process, which overcomes the limitations of lacking a deep reasoning process in the predictions. The large number of rules may result in niche and non-intuitive rules, which hinders human interpretability.

On the other hand, many Neural-Symbolic (NeSy) methods for KG reasoning, which explicitly exploit logic formulas, have been proposed in the literature (Zeng et al., 2023; DeLong et al., 2024). Some models learn logical formulas by using a neural module (Qu et al.; Cheng et al., 2023), others inject logic formulas as constraints to satisfy into the learning problem (Guo et al., 2016; Marra et al., 2020), and others inject the logic knowledge directly into an embedding representation, e.g. of a KGE (Guo et al., 2018; Marra et al., 2025). Among NeSy models, Relational Concept Bottleneck Models (R-CBMs) (Barbiero et al., 2024) have been recently designed to provide interpretable predictions in terms of human-understandable concepts, and to merge the ideas behind Concept-based models (Koh et al., 2020) with message-passing Graph Neural Networks (Gilmer et al., 2020). Despite being more interpretable, NeSy methods often focus on accuracy, while still failing to achieve the state-of-the-art in many KG tasks[1] and their evaluations primarily neglect the critical assessment of explanation quality.

In this paper, we take a different perspective to bridge the gap between high performance and interpretability in link prediction over KGs. In particular, we define a class of fully interpretable NeSy models within R-CBMs. These methods are used post-hoc, applied on top of pre-trained KGE models, ensuring both high fidelity to the selected KGE and the inherent interpretability of NeSy approaches. Our framework employs NeSy reasoners that generate explicit logic proofs using predefined or learned rules, providing transparent and explainable predictions. We evaluate the quality of these explanations using both accuracy and explainability-based metrics.

The paper is organized as follows. Section 2 introduces the background, while the interpretable NeSy models are introduced in Section 3. Section 4 reports our experimental analysis and finally Section 5 draws some conclusions and sketches possible future directions.

## 2. Background

**First-Order Logic.** We define a function-free First-Order Logic (FOL) language with a finite set of constants (domain entities) $\mathcal{C}$, variables $\mathcal{X}$ (anonymous entities), and predicates

---

1. See e.g. on FB15k-237 https://paperswithcode.com/sota/link-prediction-on-fb15k-237?metric=MRR and WN18RR https://paperswithcode.com/sota/link-prediction-on-wn18rr?metric=MRR.

(relations) $\mathcal{P}$. An atom $p(t_1, \ldots, t_n)$ consists of a predicate $p \in \mathcal{P}$ applied to constants and variables $t_1, \ldots, t_n \in \mathcal{C} \cup \mathcal{X}$; if all terms are constants, it is a *ground atom*. Logic rules use standard logical connectives $\{\neg, \wedge, \vee, \rightarrow\}$ and quantifiers $\{\forall, \exists\}$. A formula consisting of a single implication between a conjunction of atoms (the body) and another atom (the head) is a definite Horn clause. A logic theory $\mathcal{T}$ consists of rules $r_i(X_i)$, meant as universally quantified over a finite subset of variables $X_i \subset \mathcal{X}$. A substitution $\theta$ replaces variables in $X_i$ with constants from $\mathcal{C}$, yielding a *ground rule*. Grounding a rule $r_i(X_i)$ applies all substitutions, forming a set $R_i$ with $|R_i| = |\mathcal{C}|^{|X_i|}$. Grounding $\mathcal{T}$ produces $R$, the set of all ground rules. In the following, in the same spirit of methods like Markov Logic Networks Richardson and Domingos (2006), we use a Grounded Markov Network (GMN) to represent $R$ as a graph. Indeed, the graph representation of $R$ can be built by taking a node for each ground atom, and edges connecting them only if the atoms appear together in some ground formula.

**Knowledge Graph Embeddings.** Knowledge Graphs represent knowledge as triples of entities and relations, forming a graph structure (Wang et al., 2017; Dai et al., 2020). KGs are inherently incomplete, and Knowledge Graph Embeddings (Wang et al., 2021; Rossi et al., 2021) are a powerful approach, enabling inference of missing facts by mapping entities and relations to latent vectors. All KGE methods first reconstruct a fact representation from the embeddings of the entities and relations, and then compute a score from this representation via a scoring function. Here are some examples for an atom $A = p(a, b)$, where we indicate as $e_\iota$ the embedding associated to $\iota$, being either a constant, predicate, or atom:

- *TransE* (Bordes et al., 2013) models relations as translation on the embeddings of the entities, defining the atom embedding $e_A = e_a + e_p - e_b$ and its score as $1/(1 + \|e_A\|)$.

- *DistMult* (Yang et al., 2015) computes the dot product among entities and relation embeddings, defining the atom embedding $e_A = e_a \cdot e_b \cdot e_p$, being $\cdot$ the Hadamard product and its score as $\sigma(\sum_i (e_A)_i)$, being $\sigma$ the sigmoid function.

- *ComplEx* (Trouillon et al., 2016) uses the Hermitian dot product over complex embeddings to get the atom embedding, and with score $Re(\langle e_a, e_p, \bar{e}_b \rangle)$.

- *RotatE* (Sun et al.) models relations as rotations in the complex embedding space of the entities. The atom representation $e_A = e_a \circ e_p - e_b$ is scored as $-\|e_A\|$, being $\circ$ the Hadamard product and $|p_i| = 1$.

- *ExpressivE* (Pavlović and Sallinger, 2023b) embeds entity pairs as points and relations as hyper-parallelograms in a high-dimensional space. The score of each triplet is determined by assessing how well the entity pair embeddings fits within the relation's hyper-parallelogram.[2]

KGE models have demonstrated a remarkable ability to implicitly capture various logical inference patterns within their embedding representations (Pavlović and Sallinger, 2023a;

---

2. For the complete definition of the scoring function, we refer the interested reader to Equation (3) of the original paper Pavlović and Sallinger (2023b).

Table 1: Logical Rule Support by Different KGEs (cf. Pavlović and Sallinger (2023a)).

| Logical Rule | ExprE | RotatE | TransE | DistMult | ComplEx |
|---|---|---|---|---|---|
| Symmetry: $p_1(X,Y) \rightarrow p_1(Y,X)$ | ✓ | ✓ | ✗ | ✓ | ✓ |
| Anti-symmetry: $p_1(X,Y) \rightarrow \neg p_1(Y,X)$ | ✓ | ✓ | ✓ | ✗ | ✓ |
| Inversion: $p_1(X,Y) \leftrightarrow p_2(Y,X)$ | ✓ | ✓ | ✓ | ✗ | ✓ |
| Compositional definition: $p_1(X,Y) \wedge p_2(Y,Z) \leftrightarrow p_3(X,Z)$ | ✓ | ✓ | ✓ | ✗ | ✗ |
| General compositional: $p_1(X,Y) \wedge p_2(Y,Z) \rightarrow p_3(X,Z)$ | ✓ | ✗ | ✗ | ✗ | ✗ |
| Hierarchy: $p_1(X,Y) \rightarrow p_2(X,Y)$ | ✓ | ✗ | ✗ | ✓ | ✓ |
| Intersection: $p_1(X,Y) \wedge p_2(X,Y) \rightarrow p_3(X,Y)$ | ✓ | ✓ | ✓ | ✗ | ✗ |
| Mutual exclusion: $p_1(X,Y) \leftrightarrow \neg p_2(X,Y)$ | ✓ | ✓ | ✓ | ✓ | ✓ |

Abboud et al., 2020). However, as shown in Table 1, a comparison of different KGE models reveals varying degrees of success in explicitly capturing prominent logical inference patterns. It is clear that the inductive biases of modern KGEs have the potential to approximate complex logical inference patterns, even if with no direct insights into the specific nature of the inference being performed.

**Relational Concept Bottleneck Models.** Relational Concept Bottleneck Models (R-CBMs) (Barbiero et al., 2024) have been recently proposed as a novel family of methods that combine Graph Neural Networks (GNNs) (Wu et al., 2020) and concept-based XAI (Koh et al., 2020) on relational data. Assume we are given a logic theory $\mathcal{T}$ of definite Horn clause, with $R$ being the set of all ground rules. Then, for each ground atom $A = p(a,b)$, the R-CBMs' pipeline can be summarized in three steps: (i) initial atom encoding and prediction, (ii) message-passing on the GMN associated to $R$, (iii) aggregations.

(i) $A$ is encoded as $h^0(A) = g_p(\mathbf{e}_a, \mathbf{e}_b) \in \mathbb{R}^H$, with initial prediction $y^0(A) = s(h^0(A))$, being $g_p$ and $s$ the embedding and scoring functions, respectively, of some KGE, and $H$ the embedding size.

For $T > 0$ steps, with $0 < t \leq T$, the step (ii) is repeated for each ground rule $r \in R$ having $A$ as head atom, where we use the symbol $\mathcal{N}_r(A)$ to refer to the set of ground atoms that are in the body of $r$.

(ii) Message-passing updates embeddings and predictions:

$$h_r^t(A) = u_{l(r)} \left( h^{t-1}(A), \left[ h^{t-1}(B) \right]_{B \in \mathcal{N}_r(A)} \right)$$
$$y_r^t(A) = f_{l(r)} \left( y^{t-1}(A), \left[ h_r^t(B), y^{t-1}(B) \right]_{B \in \mathcal{N}_r(A)} \right)$$

being $u_{l(r)}, f_{l(r)}$ rule-type dependent functions.

(iii) Aggregation among ground rules is performed by using an aggregator operator $\oplus$, like e.g. the *max* or the *sum*:

$$h^t(A) = \sum r \in \mathcal{R}(A) h_r^t(A)$$
$$y^t(A) = \bigoplus_{r \in \mathcal{R}(A)} y_r^t(A)$$

The final embedding $h(A) = h^T(A)$ and prediction $y(A) = y^T(A)$ for each atom $A$ is obtained when $t = T$.

## 3. Models

The interpretability of standard R-CBMs is limited by complex aggregators of rule-specific predictions, $y_r^t(A)$, for atom $A$, and the black-box nature of latent representations, $h_r^t$, used in calculating $y_r^t(A)$. To address this, we propose a new class of R-CBMs that emulate logical reasoning, enabling human-understandable explanations. Our approach factorizes the computation into a rule generation or weighting phase, potentially leveraging latent features, and an interpretable rule execution phase, where the rule is symbolically executed.

We propose different models, which can be used as surrogate models to distill the output of a KGE, providing different trade-offs between expressivity and interpretability.

### 3.1. Interpretable R-CBMs

In this class of models the latent representations are used only to weight the effect of each rule, but the rule is executed symbolically in order to preserve interpretability. The message passing takes the following form:

$$h^t(A) = u\left(h^{t-1}(A), \left[h^{t-1}(B)\right]_{B \in \mathcal{N}_r(A)}\right) \tag{1}$$

$$y_r^t(A) = \overbrace{\lambda_r\left(h^t(A)\right)}^{\text{Gate}} \cdot \overbrace{tnorm\left(\left[y^{t-1}(B)\right]_{B \in \mathcal{N}_r(A)}\right)}^{\text{Semantic Rule Execution}} \tag{2}$$

$$y^t(A) = \max_{r \in \mathcal{R}(A)} y_r^t(A) , \tag{3}$$

where the $h, y$ are initialized using the KGE embeddings and outputs, respectively: $y^0(\alpha) = kge\_out(\alpha)$, $h^0(\alpha) = kge\_emb(\alpha)$ and there are $t$, with $0 \leq t \leq T$, reasoning hops by applying the rules in $R$. The idea of the proposed architecture is to structure the computation into a stage where a (non-interpretable) latent representation is computed by Equation 1. Equation 2 is used to symbolically execute the rules by performing forward chaining using the semantics of the selected tnorm, and the output is weighted by the gate in Equation 2. The selection of the form of the weighting function $\lambda_r(\cdot)$ defines different models, as detailed in the following paragraphs.

The use of maximum aggregation in Equation 3 can be seen as a soft-version of the Prolog semantics where a conclusion is true iff there is at least one supporting premise. Like in Prolog, the selection of the maximum aggregation is fundamental to preserve interpretability, because the computation of the prediction $y^t(\alpha)$ can be unambiguously traced back from the head node $\alpha$ to the supporting body nodes, and this process can be recursively repeated providing a proof tree which can be provided as an explanation. In the experimental section, we show different examples of proof trees extracted using this methodology.

**Gated R-CBMs.** This method restricts the rule weight to be a single per-rule trainable scalar $\lambda_r(h_r^t(A)) = \lambda_r^\star$. This resulting model will be referred to as Gated R-CBM (G-RCBM) and in the experimental section, we show different examples of proof trees extracted using this methodology. A Gated R-CBM has connections with other models in the literature, like Deep Logic Models (DLM) (Marra et al., 2019), which assign a weight to each rule, trained to maximize the log-likelihood of the training assignments, or Knowledge Enhanced Neural Networks (KENN) (Daniele and Serafini, 2019), which extend Semantic

Based Regularization (Diligenti et al., 2017) and Logic Tensor Networks (Badreddine et al., 2022) with a per-rule weight. These methods were originally limited to unary predicates and to one-hop reasoning for KENNs, fundamental limitations that we overcome in our formulation.

**Contextual R-CBMs (C-RCBM)**  If the rule relevance is determined based on the latent representation of a head atom, the model gains in expressiveness as it can decide that a rule contextually applies for a head atom but not another. A C-RCBM parametrizes the rule weighting by means of a neural network with a sigmoidal output such that $\lambda_r(h^t(A)) = nn_r(h^t(A))$. This model has connections with other models like Neural Markov Logic Networks (NMLN) (Marra and Kuželka, 2021). However, whereas an NMLN defines a full probability distribution over the space of assignments, the proposed R-CBMs directly compute the Maximum A-Posteriori (MAP) assignment given the training labels. This allows the method to scale to much larger relational domains.

### 3.2. Interpretable Deep Concept Reasoners (I-DCR)

DCR (Barbiero et al., 2023) is a concept-based model, which learns a formula for each head atom, given a set of candidate body atoms, then computes the output by using a tnorm:

$$
\begin{aligned}
y_r^t(A) &= tnorm\left(\left[\Phi_r(h^0(B), y^{t-1}(B))\right]_{B \in \mathcal{N}_r(A)}\right) \\
y^t(A) &= \max_{r \in \mathcal{R}(A)} y_r^t(A)
\end{aligned}
$$

where the $y_r$ are initialized using $y^0(\alpha) = kge\_out(\alpha)$, and $\Phi_r : \mathbb{R}^{H+1} \to [0,1]$ represents a logic formula processing the embedding representation and prediction of each atom in each rule $r$, to get a learned Horn Clause. In the original formulation, DCR was defined for a single step of propagation, however, we extend DCR to multiple iterations $t$, with $0 \le t \le T$, to enable multi-hop reasoning and restrict it to a max aggregation operator to merge the information from different rules. The resulting architecture, called I-DCR, can take advantage of latent representations to discover the rules to apply in a given context, unlike G-RCBMs and C-RCBMs which assume the rules to be predefined. I-DCR also preserves full human interpretability, as the generated rules are executed symbolically. The use of tnorms to perform the logic reasoning step allows an end-to-end optimization of the kge layer.

**Training.**  The different proposed models are trained to regress the values of teacher kge model using Mean Squared Error loss:

$$
L(w) = \frac{1}{2} \sum_{q \in Q \cup \tilde{Q}} \left(y_w(q) - kge\_out(h_q, r_q, t_q)\right)^2
$$

where $Q = [q_1 = (h_1, r_1, t_1), q_1 = (h_2, r_2, t_2), \ldots]$ is a set of query triples and $\tilde{Q}$ is the negative sample of query corruptions like in standard kge training and $y_w(q)$ is the output of the student model with parameters $w$.

## 4. Experimental Results

The main goal of the experiments is to compare the distilled NeSy models, namely I-DCR, G-RCBM, and C-RCBM, against different KGE models. We use as teachers ComplEx, RotatE and DistMult.

### 4.1. Datasets.

The study utilizes three datasets to evaluate reasoning capabilities. Countries (Bouchard et al., 2015) tests geographical inference across varying complexities (S1, S2, S3) by predicting country locations based on regional and neighborhood relationships. Family (Cheng et al., 2023) is designed to model transitive and hierarchical familial structures. WN18RR (Dettmers et al., 2018) is a WordNet-derived dataset specifically curated to address inverse relation leakage in knowledge graphs. A concise summary of the key statistics for each dataset is provided in Table 2.

Table 2: Detailed statistics of the datasets employed in our experiments.

| Dataset | #Entities | #Relations | #Facts | Avg. Degree | #Rules |
|---|---|---|---|---|---|
| Countries S1 | 272 | 3 | 1,110 | 4.28 | 1 |
| Countries S2 | 272 | 4 | 1,062 | 4.35 | 2 |
| Countries S3 | 272 | 4 | 978 | 4.35 | 3 |
| Family | 3007 | 12 | 19,845 | 6.47 | 48 |
| WN18RR | 40,559 | 11 | 86,835 | 2.14 | 28 |

**Rules.** The Countries Dataset defines the rules $R1 : LocIn(X, W) \wedge LocIn(W, Z) \rightarrow LocIn(X, Z)$, $R2 : NeighOf(X, Y) \wedge LocIn(Y, Z) \rightarrow LocIn(X, Z)$, $R3 : NeighOf(X, Y) \wedge NeighOf(Y, K) \wedge LocIn(K, Z) \rightarrow LocIn(X, Z)$, where $LocIn(X, Y)$ indicates that entity $X$ is located within entity $Y$, and $NeighOf(X, Y)$ indicates that entity $X$ is a neighbor of entity $Y$. Task S1 can be solved exactly with rule R1 by using information from subcontinents, Task S2 incorporates R1 and R2, it needs to use neighbor relationships to be solved, and Task S3 uses all three rules (e.g. $NeighOf(spain, france) \wedge NeighOf(france, italy) \wedge LocIn(italy, europe) \rightarrow LocIn(spain, europe)$).

For the Family and WN18RR datasets, where explicit rule sets are not predefined, we applied the AMIE (Association Rule Mining for Incomplete knowledge graphs) system (Galárraga et al., 2015). We selected a small set of top-ranking rules extracted by AMIE based on their confidence scores. The number of extracted rules for each dataset is summarized in Table 2.

### 4.2. Evaluation Metrics

To assess model performance, we employ three key evaluation metrics: Coherence, Mean Reciprocal Rank (MRR), and Hits@N.

**Coherence** plays a crucial role in knowledge distillation by evaluating the agreement between two models. It measures the proportion of queries where both models produce the

same top-ranked prediction:

$$\text{Coherence} = \frac{1}{|Q|} \sum_{q \in Q} I(\text{top prediction}_{\text{model}_1} = \text{top prediction}_{\text{model}_2}), \qquad (4)$$

where $I(\cdot)$ is an indicator function determining whether the top predictions from both models match. Higher coherence indicates that the student model successfully mimics the teacher model in knowledge distillation.

**Mean Reciprocal Rank (MRR)** quantifies the ranking quality by averaging the reciprocal rank of the first correct answer across all queries: MRR $= \frac{1}{|Q|} \sum_{q \in Q} \frac{1}{\text{rank}_q}$, where $Q$ represents the set of queries, and $\text{rank}_q$ denotes the position of the first correct answer.

**Hits@N** measures the fraction of queries for which at least one correct answer is among the top-$N$ predictions: Hits@N $= \frac{1}{|Q|} \sum_{q \in Q} I(\text{correct answer in top-}N)$, where $I(\cdot)$ is an indicator function that returns 1 if the condition is met and 0 otherwise.

### 4.3. Results

We evaluate our distillation framework on four benchmarks: Countries S2, Countries S3, Family, and WN18RR. Table 3 reports the Mean Reciprocal Rank (MRR) and Hits@1, 3, 10 for the base Knowledge Graph Embedding (KGE) models (i.e., ComplEx, RotatE, and DistMult) alongside the distilled Neural-Symbolic (NeSy) models (denoted as I-DCR, G-RCBM, and C-RCBM). [3] Please note that the reported KGE results are for the teacher models (embedding_size=100, num_train_corruptions=2), and not the best possible results obtainable for that KGE with hyperparameter tuning.

On Countries S2, the KGEs already achieve very high performance (MRRs above 0.97 for ComplEx and RotatE). Notably, the I-DCR, G-RCBM, and C-RCBM variants preserve these results. For countries S3, the baseline KGEs show a wider range of performance (with ComplEx at 0.866, RotatE at 0.956, and DistMult at 0.764). Here, the distillation process via I-DCR notably boosts the weaker DistMult baseline, while the G-RCBM and C-RCBM methods achieve high MRRs (up to 0.979) that are comparable to or better than their KGE counterparts. These results indicate that our framework can harmonize and, in some cases, enhance the relational reasoning capabilities of diverse KGEs. This could happen due to effects of, e.g., regularization or overparameterization of the teacher.

The Family dataset highlights an interesting effect of distillation. While the original KGEs yield heterogeneous performance (with ComplEx reaching 0.787 versus around 0.600 for RotatE and DistMult), the distilled models converge to nearly identical performance (around 0.764 MRR for all variants). This suggests that the distillation process may regularize the learned representations, leading to a more uniform behavior across different underlying embeddings, which can be beneficial for interpretability and downstream symbolic reasoning. On WN18RR, the baseline KGEs achieve similar performance (MRRs around 0.38), and the distillation process preserves these figures. All distilled models—across the different methods—remain close to the original performance. This consistency confirms that our approach does not compromise predictive accuracy on more challenging datasets, while still potentially offering the additional benefits of symbolic reasoning.

---

3. The code will be publicly available after paper acceptanceis available at https://github.com/rodrigo-castellano/KGE-Distillation

Table 3: MRR and Hits for the datasets Countries S2 and S3, Family and WN18RR for the different NeSy models and KGEs they are distilled from.

| Model | MRR | H@1 | H@3 | H@10 | Model | MRR | H@1 | H@3 | H@10 | Model | MRR | H@1 | H@3 | H@10 |
|---|---|---|---|---|---|---|---|---|---|---|---|---|---|---|
| | | | | | | | Countries S2 | | | | | | | |
| ComplEx | 0.979 | 0.954 | 0.996 | 1.0 | RotatE | 0.992 | 0.988 | 0.992 | 1.0 | DistMult | 0.764 | 0.764 | 0.764 | 1.0 |
| I-DCR | 0.989 | 0.979 | 0.988 | 1.0 | I-DCR | 0.982 | 0.979 | 0.996 | 1.0 | I-DCR | 0.983 | 0.967 | 0.996 | 1.0 |
| G-RCBM | 0.972 | 0.954 | 0.992 | 1.0 | G-RCBM | 0.979 | 0.962 | 0.992 | 1.0 | G-RCBM | 0.976 | 0.954 | 0.996 | 1.0 |
| C-RCBM | 0.986 | 0.975 | 0.992 | 1.0 | C-RCBM | 0.982 | 0.962 | 0.992 | 1.0 | C-RCBM | 0.981 | 0.971 | 0.992 | 1.0 |
| | | | | | | | Countries S3 | | | | | | | |
| ComplEx | 0.866 | 0.808 | 0.879 | 1.0 | RotatE | 0.956 | 0.921 | 0.992 | 1.0 | DistMult | 0.764 | 0.764 | 0.764 | 1.0 |
| I-DCR | 0.913 | 0.875 | 0.983 | 1.0 | I-DCR | 0.910 | 0.854 | 0.971 | 1.0 | I-DCR | 0.917 | 0.875 | 0.975 | 1.0 |
| G-RCBM | 0.953 | 0.908 | 1.0 | 1.0 | G-RCBM | 0.979 | 0.958 | 1.0 | 1.0 | G-RCBM | 0.962 | 0.925 | 1.0 | 1.0 |
| C-RCBM | 0.958 | 0.917 | 1.0 | 1.0 | C-RCBM | 0.979 | 0.958 | 1.0 | 1.0 | C-RCBM | 0.972 | 0.946 | 1.0 | 1.0 |
| | | | | | | | Family | | | | | | | |
| ComplEx | 0.787 | 0.653 | 0.916 | 0.951 | RotatE | 0.601 | 0.385 | 0.775 | 0.971 | DistMult | 0.609 | 0.438 | 0.735 | 0.912 |
| I-DCR | 0.764 | 0.764 | 0.764 | 0.764 | I-DCR | 0.764 | 0.764 | 0.764 | 0.764 | I-DCR | 0.765 | 0.764 | 0.764 | 0.764 |
| G-RCBM | 0.764 | 0.764 | 0.764 | 0.764 | G-RCBM | 0.764 | 0.764 | 0.764 | 0.764 | G-RCBM | 0.765 | 0.764 | 0.764 | 0.764 |
| C-RCBM | 0.764 | 0.764 | 0.764 | 0.764 | C-RCBM | 0.765 | 0.764 | 0.764 | 0.764 | C-RCBM | 0.765 | 0.764 | 0.764 | 0.764 |
| | | | | | | | WN18RR | | | | | | | |
| ComplEx | 0.384 | 0.376 | 0.387 | 0.397 | RotatE | 0.417 | 0.338 | 0.458 | 0.561 | DistMult | 0.382 | 0.370 | 0.388 | 0.399 |
| I-DCR | 0.380 | 0.366 | 0.389 | 0.404 | I-DCR | 0.381 | 0.366 | 0.392 | 0.406 | I-DCR | 0.380 | 0.366 | 0.390 | 0.405 |
| G-RCBM | 0.380 | 0.366 | 0.389 | 0.404 | G-RCBM | 0.381 | 0.366 | 0.392 | 0.406 | G-RCBM | 0.381 | 0.368 | 0.390 | 0.405 |
| C-RCBM | 0.387 | 0.375 | 0.396 | 0.407 | C-RCBM | 0.383 | 0.369 | 0.394 | 0.407 | C-RCBM | 0.380 | 0.367 | 0.390 | 0.405 |

Table 4: Coherence for the student NeSy models with the teacher models.

| | Coherence | | | | | | | | | | | |
|---|---|---|---|---|---|---|---|---|---|---|---|---|
| | ComplEx | | | | RotatE | | | | DistMult | | | |
| Model | S2 | S3 | Family | WN18RR | Model | S2 | S3 | Family | WN18RR | Model | S2 | S3 | Family | WN18RR |
| I-DCR | 0.954 | 0.796 | 0.654 | 0.367 | I-DCR | 0.979 | 0.908 | 0.385 | 0.374 | I-DCR | 0.941 | 0.808 | 0.438 | 0.365 |
| G-RCBM | 0.946 | 0.800 | 0.654 | 0.364 | G-RCBM | 0.975 | 0.904 | 0.385 | 0.373 | G-RCBM | 0.946 | 0.808 | 0.438 | 0.362 |
| C-RCBM | 0.934 | 0.796 | 0.654 | 0.369 | C-RCBM | 0.975 | 0.912 | 0.385 | 0.377 | C-RCBM | 0.930 | 0.804 | 0.438 | 0.364 |

Overall, these experiments demonstrate that our distillation framework is effective in transferring relational knowledge from high-performing KGEs to NeSy models.

**Coherence results.** Table 4 reports the coherence of the distilled NeSy models with respect to the ComplEx, RotatE, and DistMult teachers on four datasets. For Countries S2, I-DCR, G-RCBM, and C-RCBM achieve high coherence scores — 0.954, 0.946, and 0.934 respectively for ComplEx — indicating a strong alignment with the relational structure captured by ComplEx. A similar trend is observed on Countries S3, where I-DCR, G-RCBM, and C-RCBM maintain coherence scores around 0.796, 0.800, and 0.796.

On the Family dataset, all models exhibit nearly identical coherence for the same teacher, implying that the distillation process leads to a uniform alignment with the teacher in this domain. For WN18RR, coherence scores are uniformly lower (ranging from 0.364 to 0.377),

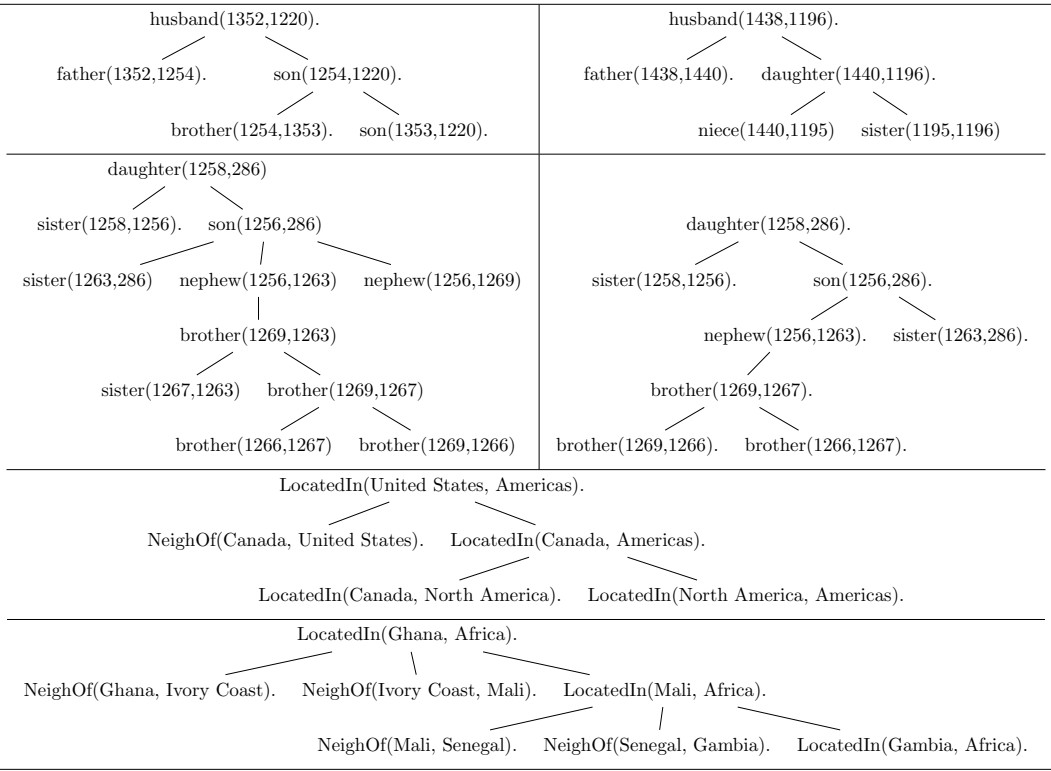

Figure 1: Examples of local explanations as proof trees obtained for the Family and Countries datasets (root nodes are test queries).

due to some atoms missing grounded rules in the NeSy methods. Therefore, coherence would have been higher by selecting a larger rule set.

**Explanations.** By tracing back the predictions, the extracted explanations represent the deep logical reasoning performed by the models. Rather than producing simple linear chains, the system constructs proof trees of varying depths by utilizing first-order logic formulas. In practice, building a proof tree from a set of straightforward rules proves to be both clearer and more accessible than forming extended flat rules. Figure 1 illustrates several explanation instances for the Countries and Family dataset, presented as proof trees. These examples highlight how sequential reasoning steps arrive at a final conclusion.

## 5. Conclusions and future work

We have presented a methodology that effectively bridges the gap between the high performance of KGEs and the need for interpretable reasoning in link prediction tasks. By distilling the knowledge captured by KGEs into a neural-symbolic framework, we generate detailed logical proofs, offering a level of explanation far beyond the capabilities of the original black-box models. Future research will explore the refinement of this distillation process to handle more complex knowledge graphs and rule structures.

## Acknowledgments

This work has been partially supported by the project PNRR M4C2 "FAIR - Future Artificial Intelligence Research" - Spoke 1 "Human-centered AI" , code PE0000013, CUP E53C22001610006. This work was also supported by the EU Framework Program for Research and Innovation Horizon under the Grant Agreement No 101073307 (MSCA-DN LeMuR). This work has been partially supported by the project "CONSTR: a COllectionless-based Neuro-Symbolic Theory for learning and Reasoning", PARTENARIATO ESTESO "Future Artificial Intelligence Research - FAIR", SPOKE 1 "Human-Centered AI" Università di Pisa, "NextGenerationEU", CUP I53C22001380006.

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
