# OpenReview forum: "Distilling KGE black boxes into interpretable NeSy models"
_nesyconf.org/NeSy/2025/Conference — NeSy 2025 Poster_

### Official Review · Reviewer_SQbb · 2025-03-28
**Interesting approach, but technical clarity needs to be polished**

**Rating:** 6
**Confidence:** 4

**Review:**

**Summary**

This paper presents a novel method for link prediction relying on knowledge graph embeddings. The method aims to be both highly accurate and interpretable. The method is based on the application of a Graph Neural Network on the initial knowledge graph embedding, where each layer of the network is designed so that it represents a weighted application of a known rule, which may or may not be applied depending on the specific constants that ground it. In this way, each prediction by the model can be seen as the result of the serial application of several rules. The authors show that their approach provides a performance comparable or better than other state-of-the-art approaches, with the added benefit that each prediction can be explained by a rule.

**Strengths**

 - *Performance.* The experiments show that the approach presented has an accuracy performance similar or superior to state-of-the-art approaches.

 - *Originality.* The approach presented by the paper is novel, although similar to existing approaches. It addresses an important question: how to achieve high accuracy in link prediction while maintaining interpretability.


**Weaknesses**

 - *Technical clarity.* The paper is somewhat lacking in technical clarity. There is no precise definition of the approach presented, as important concepts are not defined, and crucial questions remain about the role of the rules in the model.

 - *Significance.* I am not persuaded that the method provided is a scalable approach to guarantee interpretability, because it relies on a rather non-standard and non-general application of logical rules.

- *Motivation.* The motivation presented for the approach is weak, especially when compared with rule-learning approaches.


**Resolution**

The suggested approach is likely to be of interest to the NeSy community, and so I think this warrants acceptance. However, I would suggest that the paper is revised to address the outstanding technical issues, improve its clarity, and make it more self-contained.

**Detailed Review**

*Motivation and Contribution*

I am not persuaded that the presented approach provides better interpretability than existing methods. First, rule-mining methods like AMIE already provide short and readable rules, which are easy to interpret (certainly more interpretable than rules with complex, non-standard semantics). The model presented in this paper uses a rather non-standard application of rules, where rules are triggered or not depending on the specific identities of the constants that unify the variables. I find this highly counterintuitive: the advantage of rules is precisely in their generality. If I can extract rules that seem to be universally quantified, but the model only applies these rules to some constants (and I can't know which constants, because this depends on the black-box embeddings of the model) I'd argue that the rules do not help me truly understand how the model works. Perhaps I have misunderstood something?

*Literature Review*

There is a thorough discussion of relevant literature, although I was surprised to see that neural rule-learning models, especially those also based on graph neural networks, were not discussed. There exist already GNN (and non-GNN)-based models where one can extract interpretable rules with very clear semantics that are faithful to the model. These approaches seem highly relevant to the paper.

For the sake of transparency, it is also worth pointing out that this approach only works (seemingly) for the transductive link prediction setting, where the constant set is fixed and finite.

*Technical clarity*

Technical clarity could be improved. For example, the tnorm function could be defined, especially since in eq (2) the argument seems to be a vector, but in the equation in Section 3.2 it seems to be a Boolean value. Many important symbols are not defined, for example, Phi_r in Section 3.2, which makes understanding the approach quite challenging. It would also be important to be able to see a clear syntax and semantics for the extracted rules; instead, the paper only informally discusses how such syntax and semantics resembles and differs from other approaches such as Neural Markov Logic Networks or Logic Tensor Networks.

In Section 3.1, could you clarify how to interpret the feature vectors for each vertex, and the application of a rule as a T-norm?

Here are some other minor comments:

 > "A logic theory T consists of rules ri(Xi), meant as universally quantified over each X_i ⊂ X"

This is strange. There are typically infinite Xi's and quantification over all of them would yield an infinite rule. Perhaps the intention was to say each variable in X_i, for X_i in X?

 > "Grounding T produces R, the set of all ground rules. [...] we use a Grounded Markov Network (GMN) to represent R as a graph"

 this seemingly requires that the set of constants is finite, otherwise the set of all substitutions is infinite, and cannot be represented in a graph in practice. This should probably be mentioned explicitly, to avoid confusion.

 > "A is encoded as h0(A) = gp(ea,eb) ∈RH , with initial prediction y0(A) = s(h0(A)), being gp and s the embedding and scoring functions, respectively, of some KGE."

Most symbols used here are not defined.

**Anonymity:**

Disclose identity

---

### Official Review · Reviewer_emDd · 2025-04-03
**Revision of Distilling KGE black boxes into interpretable NeSy models**

**Rating:** 6
**Confidence:** 4

**Review:**

## Summary

This paper proposes a set of post-hoc models to generate explanations of link predictions in knowledge graphs using knowledge graph embeddings.  The post-hoc models are inspired on Relational Concept Bottleneck Models, which the authors claim the extend/modify to overcome interpretability issues.

## Comments

- Some entries in Table 1 may be inaccurate. For example, how can TransE encode
  Symmetry? for p(X,Y) and p(Y,X) to hold the expressions e_X + W_p =
  e_Y and e_Y + W_p = e_X should hold as well. Therefore, W_p should
  be 0, which is a rather meaningless embedding for a relation. I
  believe a similar argument can be applied to Hierarchy on
  TransE. Additionally, is it true that RotatE does not preserve
  composition? There is a similar table with different information in
  this paper: https://arxiv.org/pdf/2110.14450. Can the authors
  contrast both tables and explain the discrepancies and analyze the
  correctness of Table 1?

- In the Countries dataset the explanation of tasks S2, and S3 are too
  obscure. Is it possible to have at least an example of S2 and S3?
  That would help to understand why rules R2 and R3 are useful on
  those tasks.

- The methods in this paper rely strongly on the ones provided in
  https://arxiv.org/abs/2308.11991. However, I think this paper would
  benefit from a proper comparison to the original methods. For
  example, the authors introduce the G-RCBM and C-RCBM architectures and
  claim they overcome interpretability issues from RCBMs. The same
  claim is done for the I-DCR architecture which comes from R-DCR in
  the cited paper. My question is: how the distillation process would
  work when using the standard RCBMs and R-DCR architectures? Are
  RCBMs and R-DCR models capable of generating explanations at all? If
  RCBMs and R-DCR can generate explanations: how do those explanations
  differ from the ones generated by the models proposed here?

- In Figure 1 there are proof trees. Do they correspond to a particular
  query made on the graphs? If so, could authors explicitly show the
  queries? Additionally, Figure 1 is mistakenly referenced as Table 1
  in the "Explanations" paragraph.

- Could the authors elaborate on what metrics are used to assess the
  quality of explanations? The abstract and introduction mention the
  use of "accuracy and explainability metrics." I understand that MRR
  and Hits@K evaluate prediction accuracy. However, it is unclear
  whether the authors are using coherence as the primary
  explainability measure. While coherence captures agreement between
  models, it does not necessarily reflect whether an explanation is
  accurate, informative, or helpful to a human user.

## Minor comments
- There are some references that point to Arxiv preprints despite
  published versions exists. This should be fixed.

- There are some typos I could identify:

  - Section 3). --> Section 3.
  - whether all terms are constants -->  if all terms are constants,
  - R-CBM --> RCBM (both ways are used throughout the paper. Please unify the notation)
  - kge embeddings --> KGE embeddings
  - This could happend --> This could happen
  - effects such --> effects of?

**Anonymity:**

Remain anonymous